# Assessing Non-Laboratory Healthcare Professionals’ Attitude towards the Importance of Patient Preparation for Laboratory Tests

**DOI:** 10.3390/healthcare12100989

**Published:** 2024-05-10

**Authors:** Ričardas Stonys, Dalius Vitkus

**Affiliations:** 1Department of Physiology, Biochemistry, Microbiology and Laboratory Medicine, Institute of Biomedical Sciences, Faculty of Medicine, Vilnius University, M.K. Ciurlionio str. 21, LT-03101 Vilnius, Lithuania; dalius.vitkus@santa.lt; 2Centre of Laboratory Medicine, Vilnius University Hospital Santaros Klinikos, Santariskiu str. 2, LT-08406 Vilnius, Lithuania

**Keywords:** preanalytical phases, attitude of health personnel, clinical laboratory testing, quality of health care

## Abstract

(1) Background: Various guidelines address patient preparation and its importance for venous blood sampling, such as the GP41 guideline issued by the Clinical Laboratory Standards Institute (CLSI) and the blood collection guidelines published by the World Health Organisation. Recommendations provided by national societies or international organisations in the field of radiology, such as The Contrast Media Safety Committee of the European Society of Urogenital Radiology, or in the field of laboratory medicine, such as the Working Group for Preanalytical Phase (WG-PRE) of the European Federation of Clinical Chemistry and Laboratory Medicine (EFLM) and the Latin American Working Group for Preanalytical Phase (WG-PRE-LATAM) of the Latin American Confederation of Clinical Biochemistry (COLABIOCLI), also guide this practice. There is a notable lack of understanding regarding the viewpoints held by non-laboratory healthcare professionals concerning the significance of patient preparation for laboratory testing and the impact of typical factors associated with patient preparation. This study endeavours to bridge this gap by assessing the attitude of non-laboratory healthcare professionals in Lithuania regarding these pivotal aspects. (2) Methods: A self-designed anonymous questionnaire was disseminated among 141 public healthcare institutions in Lithuania. The internal consistency of the questionnaire was evaluated by computing Cronbach’s alpha. Descriptive statistics were utilised for the variables, while comparisons of attitude among groups were conducted using Mann–Whitney U (for two groups) or Kruskal–Wallis (for more than two groups) for categorical and discrete indicators. The Kruskal–Wallis post-hoc test was employed for pairwise comparisons. A significance level of *p*-Value < 0.05 was applied to establish statistical significance. (3) Results: A total of 158 respondents constituted two distinct groups of healthcare professionals: nurses and physicians. Most of the participants either agreed or strongly agreed that patient preparation could introduce bias into laboratory test results. Professionals with less than 20 years of work experience or those who attended training in patient preparation for sampling within a 5-year timeframe exhibited stronger agreement regarding different preanalytical factors in patient preparation and their impact on laboratory test results compared to their counterparts. (4) Conclusions: Non-laboratory healthcare professionals who participated in this survey consider proper patient preparation for laboratory testing to be a significant step towards obtaining accurate test results. They also recognize the commonly acknowledged preanalytical factors as important for ensuring reliable test results. However, attitudes towards the importance of several preanalytical factors vary depending on whether non-laboratory healthcare professionals have more or less than 20 years of work experience, as well as whether they have attended any training on this topic within the last five years or have never attended such training.

## 1. Introduction

The significance of laboratory medicine in healthcare cannot be overstated, as it underpins clinical decision-making and patient management. The laboratory testing process encompasses various stages, commencing with test ordering and culminating in the delivery and interpretation of results [1]. Patients encounter diverse healthcare professionals during their care journey, with laboratory tests being no exception. Several factors affecting the reliability of laboratory test results manifest in the preanalytical phase, occurring outside the laboratory. These factors encompass aspects such as patient preparation before sample collection and patient identification, as well as specimen handling and storage [2].

Patient preparation plays a pivotal role in ensuring the accuracy and reliability of blood laboratory test results. Various guidelines exist that address patient preparation and its importance for venous blood sampling, for instance, the GP41 guideline issued by the Clinical Laboratory Standards Institute (CLSI) in 2017 [3] and the blood collection guidelines published by the World Health Organisation in 2010 [4]. Recommendations provided by national societies [5] or international organisations in the field of radiology, such as The Contrast Media Safety Committee of the European Society of Urogenital Radiology [6], or in the field of laboratory medicine, such as the Working Group for Preanalytical Phase (WG-PRE) of the European Federation of Clinical Chemistry and Laboratory Medicine (EFLM) and the Latin American Working Group for Preanalytical Phase (WG-PRE-LATAM) of the Latin American Confederation of Clinical Biochemistry (COLABIOCLI) [7,8], also guide this practice. Several years ago, the Working Group for Preanalytical Phase (WG-PRE) of the European Federation for Clinical Chemistry and Laboratory Medicine (EFLM) proposed that blood for all routine blood tests should be taken preferably in the morning, after an overnight fast (12 h), allowing water consumption, but prohibiting the consumption of tea, coffee, and other caffeine-containing drinks or alcohol, refraining from intensive physical activity and cigarette smoking before blood sampling [7]. These recommendations align with various studies that have shown that non-fasting [9,10], or the use of chewing gum before sampling [11], intensive physical activity [12], and medication use [13,14,15], could introduce bias into laboratory test results. Beside these factors, it is known that circadian rhythm [16], day of menstrual cycle [17,18], and sample contamination with contrast media [6,19,20] in the blood can introduce significant variability into laboratory test results [19,21], and these preanalytical aspects have to be taken into account when preparing the patient for testing as well. Patients should be required to adhere to these guidelines, which may involve fasting for particular tests or refraining from certain medications to prevent potential interference or erroneous outcomes [8]. Non-compliance with these instructions can significantly impact the quality of test results [9]. According to ISO 15189, laboratories should be responsible for preparing clear instructions on the topic of patient preparation for laboratory testing and providing them to non-laboratory healthcare professionals [22]. Non-laboratory healthcare professionals should follow these instructions, as effective interdisciplinary collaboration is fundamental at this juncture. From a legal standpoint, both laboratory and non-laboratory professionals are permitted to provide this type of information to patients in Lithuania. However, non-laboratory professionals, including physicians and nurses, are better positioned to educate patients on the importance of adequate preparation. This usually involves explaining the purpose of the laboratory test and obtaining informed consent. Current studies have shown that patients often present for laboratory tests ill-prepared [23,24,25,26]. Multiple factors contribute to this lack of preparation, including variations in preparation recommendations from different laboratories [8,27,28] and patient attitudes towards the significance of preparation for laboratory testing [29]. Additionally, the attitude of non-laboratory healthcare professionals regarding the importance of patient preparation for laboratory testing can be another influential factor affecting the quality of patient preparation. This aspect remains understudied. Evaluating the viewpoints of non-laboratory professionals contributes to a patient-centric approach, enhancing patient satisfaction and fostering trust in the healthcare system. Ultimately, assessing the attitude of non-laboratory healthcare professionals regarding the importance of patient preparation seeks to elevate the quality of patient care. Adequately prepared patients yield more precise test results, thus enhancing the accuracy of diagnosis and treatment decisions.

Currently, there is a dearth of knowledge regarding the attitude of non-laboratory healthcare professionals in Lithuania regarding the importance of patient preparation for laboratory testing and the influence of common factors associated with patient preparation. This study aims to address this gap by evaluating the viewpoints of non-laboratory healthcare professionals in Lithuania on these crucial aspects.

## 2. Materials and Methods

### 2.1. Study Design

To evaluate the general attitude regarding the importance of patient preparation significance within the entire testing process, we conducted a survey involving non-laboratory healthcare professionals, specifically nurses and physicians. This survey took place over a span from February 2023 to October 2023. The research employed a self-designed anonymous questionnaire, which was prepared by laboratory medicine professionals.

The questionnaire consisted of questions centred on the demographic profile of the participants, encompassing variables such as gender, work experience, education level, professional title, and participation in training related to patient preparation for laboratory tests. The other part of the questionnaire consisted of 12 questions related to patient preparation before venous blood sampling, each rated on a 5-point Likert scale, with responses coded from one to five, ranging from “strongly disagree” to “strongly agree”. These questions were chosen and constructed by specialists in laboratory medicine, drawing upon similar studies in this field and the available international guidelines for venous blood sampling, specifically EFLM-COLABIOCLI recommendations. Examples of the questions (translated into English from Lithuanian) are provided in Table A1.

Prior to distribution, the questionnaire underwent a reassessment to ensure its face validity, which entails evaluating whether the questions appear suitable and relevant for the intended construct and objectives. Due to the clear and linguistically adapted nature of the questions to the Lithuanian context, they were deemed understandable and pertinent. As each item was individually evaluated without contributing to a cumulative score, this level of validation was deemed sufficient to confirm the questionnaire’s appropriateness.

### 2.2. Data Collection

The questionnaire was distributed electronically using Google Forms to 141 public healthcare institutions in Lithuania. The invitation letter with the link to the questionnaire was sent via email to the public healthcare institutions using the contacts found on the webpage of the Ministry of Health of the Republic of Lithuania. These institutions were requested to share the questionnaire with their employees, specifically nurses and physicians. Respondents had to indicate their consent by clicking a button before proceeding, ensuring they were aware their responses would be used anonymously for analysis and publication. They were also provided with clear instructions and assurances regarding data confidentiality. Data were collected anonymously (no data regarding respondents’ traceability were collected, nor did the researchers know which employees were invited), obviating the need for specific informed consent and negating the requirement for Ethics Committee approval, as, under Lithuanian law, it did not fall within the legal parameters of a biomedical study. Respondents had no time constraints for completion the questionnaire and were not offered incentives for participating. It was obligatory to respond to all the questions.

### 2.3. Statistical Analysis

All responses from the survey were exported to Excel and converted into numerical values. The internal consistency of the questionnaires was assessed by calculating Cronbach’s alpha. Due to the limited number of respondents, ≤5-, 6–10-, 11–15-, and 16–20-years’ work experience groups were combined into <20 years. The groups of respondents who reported ≤1, ≤3, and ≤5 years since their last attendance at training concerning the patient preparation for laboratory test were consolidated into a single group referred to as the ≤5 years ago group. Respondents, according to their reported professional title, were divided into two groups: nurses and physicians. For statistical analysis, the responses to the 5-point Likert scale questions were converted to a scale that spans from one (representing strong disagreement) to five (indicating strong agreement). The normality of the converted Likert scale question responses was assessed using the Shapiro–Wilk test, which revealed a non-normal distribution. Descriptive statistics are presented for the variables, and comparisons of attitudes between groups (work experience, training, etc.) were performed using a nonparametric test such as Mann–Whitney U (for two groups) or Kruskal–Wallis (for more than two groups) for categorical and discrete indicators. The Kruskal–Wallis post-hoc test was employed for pairwise comparisons. A significance level of *p*-Value < 0.05 was used to determine statistical significance. The analysis was performed using the Statistical Package for Social Sciences version 22.22 (SPSS Inc., Chicago, IL, USA).

## 3. Results

The internal consistency of the questionnaire was assessed using Cronbach’s Alpha, and the results indicated good reliability (α = 0.895) for the scale, which consisted of 12 questions rated on a 5-point Likert scale.

A total of 165 responses were received over the course of this study. Following an initial data screening, seven responses were deemed ineligible for inclusion. These responses were sourced from laboratory personnel or other hospital staff members who typically do not engage in patient preparation preceding laboratory tests. The final sample of 158 respondents comprised two distinct groups of healthcare professionals: nurses and physicians. Most of the physicians were general practitioners (*n* = 18; 34.6%), anaesthesiologist-intensivists (*n* = 6; 11.5%), obstetricians-gynaecologists (*n* = 6; 11.5%), or internal medicine physicians (*n* = 7; 13.5%). Due to the limited number of physicians per professional title, no analysis regarding the professional title was conducted. Instead, all these participants were combined into one group of physicians. Among the nurses, the majority were female, consistent with the country’s workforce composition. The demographic characteristics of the participants are summarized in Table 1.

In the case of each item, the mode exhibited variability, with a conspicuous concentration of responses primarily falling within the range of agree and strongly agree. Overall, more than half of the healthcare professionals, encompassing both nurses (Q1, with 89.7% strongly agreeing or agreeing) and physicians (Q1, with 88.5% strongly agreeing or agreeing), espouse the belief that patient preparation exerts a substantial influence on the quality and reliability of test results. A somewhat larger proportion of professionals in the nurses’ group (ranging from 68.8% to 87.7%) compared to the physicians’ group strongly agreed or agreed that various factors such as the consumption of food (Q2) and beverages (Q9), alcohol (Q3), smoking (Q5), the usage of food supplements (Q7), or medication (Q8) can introduce bias into laboratory test results. It is worth noting that, for the same questions, a higher proportion of physicians expressed uncertainty or lacked a clear opinion compared to the nurses’ group. Conversely, the opposite situation is observed for the questions concerning the impact of intensive physical activity (Q6), contrast media (Q10), the day of the menstrual cycle (Q11), and circadian rhythm (Q12) on laboratory test results. In these cases, a slightly larger proportion of physicians (ranging from 65.6% to 80.7%) strongly agreed or agreed with the influence of these factors compared to the nurses’ group. In both groups of professionals, nurses and physicians, the most of respondents agreed or had no clear opinion about the notion that consuming a glass of water before laboratory tests (Q4) can introduce bias into their results. The comprehensive questionnaire outcomes for nurses and physicians are consolidated in Figure 1 and Figure 2, correspondingly.

Statistically significant differences in viewpoints regarding the impact of smoking (Q5) (*p*-Value = 0.020), intensive physical activity (Q6) (*p*-Value = 0.017), the usage of food supplements (Q7) (*p*-Value = 0.023), and the day of the menstrual cycle (Q11) (*p*-Value = 0.024) were evident among participants with less than 20 years versus more than 20 years of work experience. Notably, a greater proportion of participants with less than 20 years of experience agreed that these patient preparation factors can impact the quality of laboratory testing compared to those with over 20 years of experience. Also, significant differences were observed based on whether participants had attended training related to patient preparation for laboratory testing within the past 5 years or had never attended such training. These differences were observed solely for the overall attitude regarding the importance of patient preparation (Q1) (*p*-Value = 0.013) and the impact of smoking (Q5) (*p*-Value = 0.034), showing that a greater proportion of participants who attended training within the past 5 years exhibited stronger agreement on the importance of these factors compared to those who never attended such training. A summary of these observations is presented in Figure 3.

To examine potential discrepancies in attitude distribution among nurses and physicians based on their education level, work experience, and participation in training regarding this topic, a pairwise comparison was conducted. However, due to the small number of respondents in the physicians’ group holding a PhD degree, it was not feasible to compare viewpoints between physicians based on their level of education. In the nurses’ group, no statistically significant differences in opinions across the 12 items were detected concerning their education level or work experience. However, within the physicians’ cohort, statistically significant differences emerged only for the questions concerning the impact of intensive physical activity (Q6) (*p*-Value < 0.001) and the usage of food supplements (Q7) (*p*-Value < 0.001), indicating that a larger proportion of respondents with less than 20 years of experience agreed more on the importance of these factors (Figure 4). Regarding participation in training for patient preparation for laboratory testing, no statistically significant difference in opinions was evident among physicians and nurses.

Furthermore, we analysed variances in attitude among nurses and physicians based on their work experience and participation in training. Physicians and nurses with less than 20 years of work experience showed statistically significant differences in attitude regarding the importance of intensive physical activity (Q6) (*p*-Value = 0.010) and usage of food supplements (Q7) (*p*-Value = 0.023), with physicians expressing stronger agreement on these factors. Additionally, distinctions were noted in the usage of food supplements (Q7) (*p*-Value = 0.005) and beverages (Q9) (*p*-Value = 0.010) between nurses and physicians with over 20 years of work experience, with nurses expressing stronger agreement on these factors. Concerning participation in training, no statistically significant difference emerged in attitudes among nurses and physicians who participated in training on the topic of patient preparation within a 5-year period. The comprehensive summary of these differences is presented in Figure 5.

## 4. Discussion

Firstly, the results of the analysis of internal consistency of the questionnaire indicated a high level of reliability. This suggests that the 12-item scale, which was used to gauge participants’ opinions, served as a robust and consistent measurement tool. This study revealed several noteworthy findings and offered insights into the attitudes and opinions concerning the significance of patient preparation factors for the quality of laboratory testing among both nurses and physicians. We observed a strong consensus in the attitudes of both nurses and physicians regarding the significance of patient preparation factors frequently mentioned in the scientific literature that can impact the reliability and accuracy of laboratory test results. Study findings demonstrate that, essentially, the opinions of both physicians and nurses regarding the influence of the preanalytical variables on laboratory test results align with existing research, evidence, and guidelines. Interestingly, our data revealed that most of the nurses and physicians either disagreed or held no specific opinion (neither agreed nor disagreed) that drinking a glass of water before laboratory tests could introduce bias into results. This observed attitude contradicts current scientific evidence, which indicates that this factor has no influence on laboratory results [30,31]. Such a misleading attitude could lead to patients being instructed to follow unnecessary requirements. However, further studies should be conducted to justify these assumptions.

The data from our study revealed that respondents with less than 20 years of experience had a stronger consensus regarding the impact of several preanalytical factors in the patient preparation step. Specifically, they expressed stronger agreement on the impact of smoking, intensive physical activity, the usage of food supplements, and the day of the menstrual cycle on laboratory test results compared to their counterparts with more extensive work experience (Figure 3 and Figure 4). These disparities could arise from several conceivable factors. It is plausible that respondents with fewer than 20 years of experience are more aligned with current practices and education, benefiting from updated training and a heightened emphasis on preanalytical factors. However, it is important to note that some physicians with more than 20 years of experience may have also received additional training within the previous five years, potentially influencing their perspectives on these factors. Moreover, our findings suggest that nurses within the more than 20 years’ work experience bracket were more inclined to agree on certain patient preparation preanalytical factors influencing test results compared to their physician counterparts (Figure 5). This disparity could be attributed to nurses’ more frequent communication with the laboratory, potentially shaping their attitude based on daily interactions and information exchange. Further investigations are required to validate these assumptions.

Also, the data from our study revealed that respondents who participated in training on patient preparation were more likely to agree that patient preparation and certain factors (Figure 3) can influence the accuracy of laboratory test results. These findings align with similar studies’ results, which have shown that continuous training is a perspective tool in enhancing the quality of the preanalytical phase [32,33,34]. In our case, it may influence the perception of participants on the importance of these preanalytical factors.

While our study focused on the attitudes of non-laboratory healthcare professionals towards the importance of patient preparation for laboratory testing, it is essential to consider the broader implications of these findings. Understanding the factors that influence attitudes towards patient preparation is essential for enhancing the quality of healthcare delivery overall. Further research in this area could help to refine and improve existing training programs, ultimately leading to better patient outcomes and more reliable laboratory test results. By addressing knowledge gaps and promoting awareness of best practices in patient preparation, healthcare organizations can improve the quality of care provided to patients and enhance overall patient outcomes.

Studies have shown that only a minority of the patients tend to arrive well-prepared for blood sample collection [23,29,35], and this is no exception in Lithuania [26]. The reasons for this trend could be several. Firstly, patients could be not well informed about preparation for laboratory testing. The first source for information about laboratory tests for patients is their physicians, as studies have shown [36]. However, studies have shown that non-laboratory healthcare professionals tend to lack knowledge about various preanalytical factors, for example, ideal fasting duration [34]. This is why they may be not willing to present information on a topic they lack knowledge about. According to ISO 15189, laboratories should be responsible for training non-laboratory professionals and providing information regarding preanalytical issues [22]. Secondly, patients might not adhere to the instructions provided for them because of their attitude to the importance of various factors regarding this step [29,35]. Thirdly, as shown by several studies, laboratories tend to provide information about patient preparation for laboratory testing that is not standardized and not fully covering all the aspects [8,27,28]. This variability can lead to the point that patients are provided with different recommendations on the same topic, which leads to the fact that correct practice cannot be learned in the general population, affecting the quality of this step. The variability in the information provided to patients for preparation could also arise from the difference in attitudes about the importance of preanalytical factors, as professionals may tend to provide information to the patient regarding only the factors they believe are important.

This study acknowledges several limitations that should be taken into consideration when interpreting the findings. Firstly, relatively low sample sizes for physician-reported questionnaires limited our ability to perform inferential analysis in this subgroup by specializations. Secondly, participants may have provided answers that were not entirely sincere or accurate, which could affect the reliability of the data. Also, calculating the response rate is challenging due to the fact that not all healthcare institutions provided numbers of professionals to whom the questionnaire was shared, preventing precise determination of the total invited individuals. These limitations could introduce a potential source of selection bias and limit the generalisability of the findings to the entire population of professionals in the country. Nevertheless, our findings provide general insights into non-laboratory healthcare professionals’ attitudes on the importance of preanalytical phase variables and could stimulate further, more comprehensive studies on this topic.

## 5. Conclusions

In conclusion, non-laboratory healthcare professionals who participated in this survey consider proper patient preparation for laboratory testing to be a significant step towards obtaining accurate test results. They also recognize the commonly acknowledged preanalytical factors as important for ensuring reliable test results. However, attitudes towards the importance of several preanalytical factors vary depending on whether non-laboratory healthcare professionals have more or less than 20 years of work experience, as well as whether they have attended any training on this topic within the last five years or have never attended such training.

## Figures and Tables

**Figure 1 healthcare-12-00989-f001:**
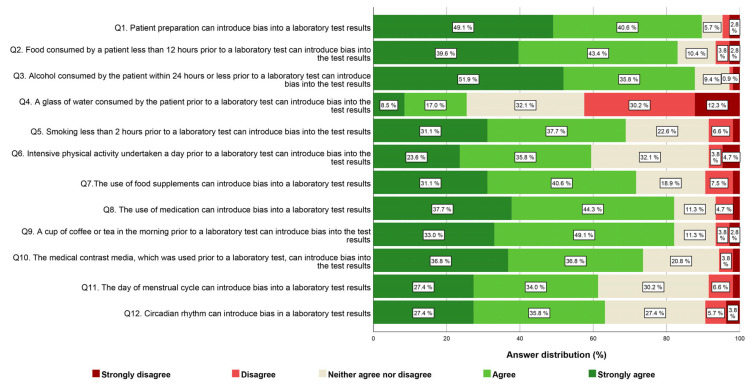
Nurse-provided responses regarding agreement on the importance of various preanalytical variables on laboratory test results, organized by question.

**Figure 2 healthcare-12-00989-f002:**
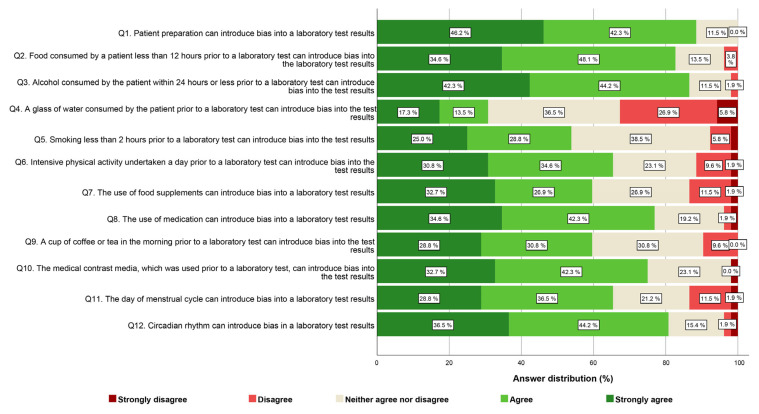
Physician-provided responses regarding agreement on the importance of various preanalytical variables on laboratory test results, organized by question.

**Figure 3 healthcare-12-00989-f003:**
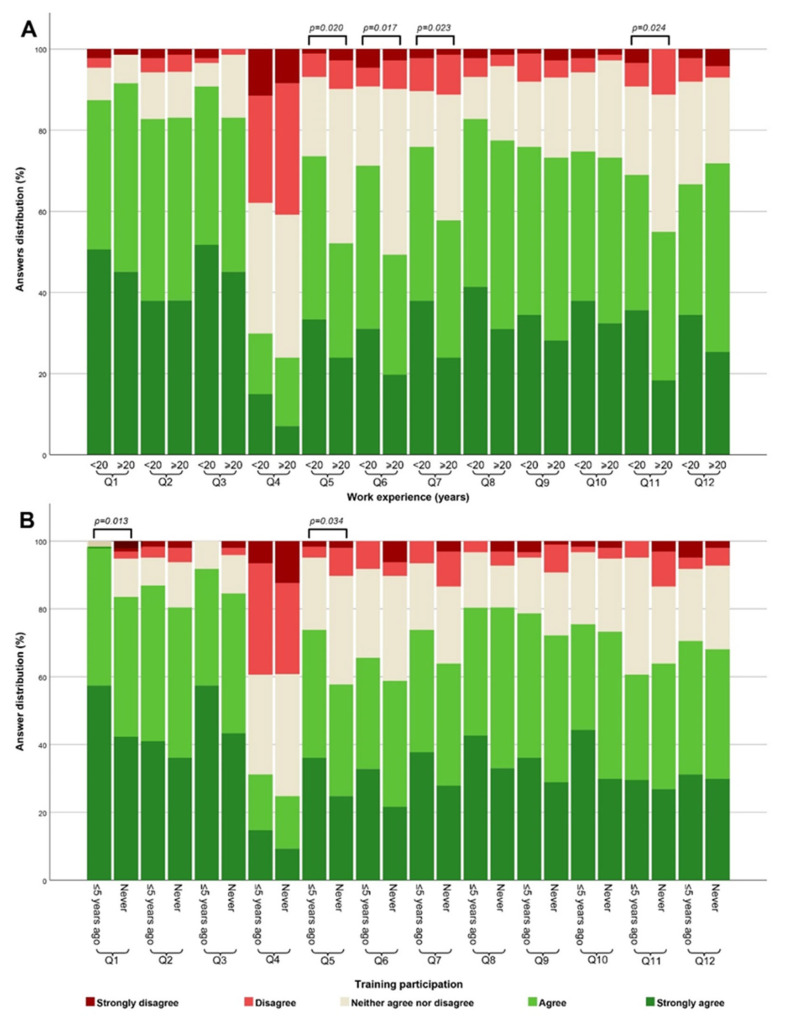
Comparison of attitudes towards the influence of various patient preparation factors on laboratory test results among different groups. Variations in attitude are presented among respondents with less than 20 years of work experience compared to those with more than 20 years (**A**), as well as between respondents who underwent training on this topic within a 5-year period and those who had never participated in such training (**B**). All *p*-Values were calculated by Mann–Whitney U test.

**Figure 4 healthcare-12-00989-f004:**
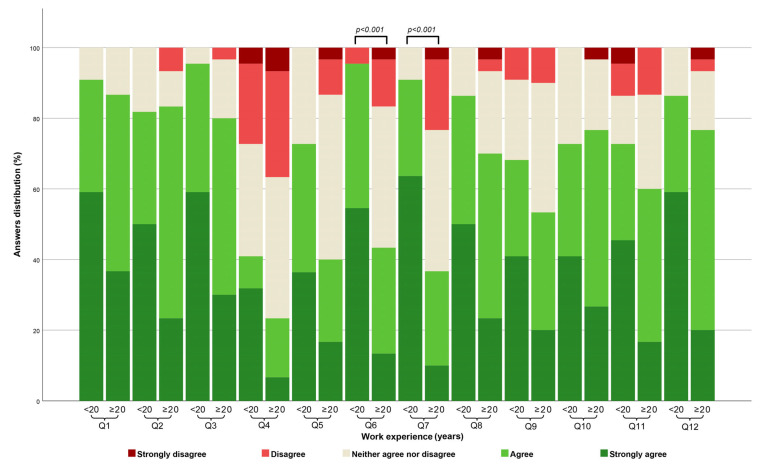
Comparison of attitude towards the influence of various patient preparation factors on laboratory test results among physicians with less than 20 years versus more than 20 years of work experience. Pairwise comparisons utilized the Kruskal–Wallis H post-hoc test. All *p*-Values are adjusted using Bonferroni correction for multiple tests.

**Figure 5 healthcare-12-00989-f005:**
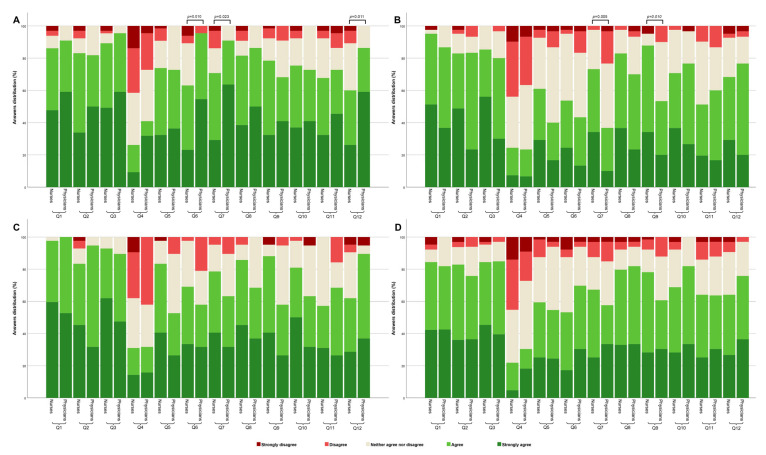
Comparison of attitudes towards the influence of various patient preparation factors on laboratory test results among nurses and physicians. The data are presented by categorizing respondents into distinct groups: those with less than 20 years of work experience (**A**), respondents with more than 20 years of work experience (**B**), respondents who attended training on patient preparation for laboratory tests within a 5-year timeframe (**C**), and those who have never attended such training (**D**). Pairwise comparisons utilized the Kruskal–Wallis H post-hoc test. All *p*-Values are adjusted using Bonferroni correction for multiple tests.

**Table 1 healthcare-12-00989-t001:** Demographic characteristics of healthcare professionals by group, gender, education, work experience, and training participation.

Characteristic	Nurses, *n* = 106 (%)	Physicians, *n* = 52 (%)
Sex		
Female	102 (96.2)	39 (75.0)
Male	4 (3.8)	13 (25.0)
Education		
Professional bachelor’s degree	59 (55.7)	0 (0)
University bachelor’s degree	25 (23.6)	0 (0)
University master’s degree	22 (20.7)	47 (90.4)
PhD degree	0 (0)	5 (9.6)
Work experience		
<20 years	65 (61.3)	22 (42.3)
≥20 years	41 (38.7)	30 (57.7)
Training participation		
Yes, ≤5 years ago	42 (39.6)	19 (36.9)
I have not participated in such training	64 (60.4)	33 (63.1)

Data are shown as number (%).

## Data Availability

The available dataset can be provided upon reasonable request to the corresponding author.

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
