# Peer review of "Assessing Non-Laboratory Healthcare Professionals’ Attitude towards the Importance of Patient Preparation for Laboratory Tests"

_healthcare, 2024, doi:10.3390/healthcare12100989_

Round 1

Reviewer 1 Report

Comments and Suggestions for Authors

A validated questionnaire must be used.

The conclusions section is missing.

Comments on the Quality of English Language

Minor editing of  english lenguage required

Author Response

Dear Reviewer,

Thank you very much for taking the time to review our manuscript titled "Assessing Non-Laboratory Healthcare Professionals' Attitude Towards the Importance of Patient Preparation for Laboratory Tests" We appreciate the thoughtful feedback and constructive comments you provided. Below, we address each of your points and outline the revisions we have made in response:

Comment 1. A validated questionnaire must be used.

Response 1. Prior to distribution, the questionnaire underwent a reassessment to ensure its face validity, which entails evaluating whether the questions appear suitable and relevant for the intended construct and objectives. Due to the clear and linguistically adapted nature of the questions to the Lithuanian context, they were deemed understandable and pertinent. As each item was individually evaluated without contributing to a cumulative score, this level of validation was deemed sufficient to confirm the questionnaire's appropriateness. This is also now documented in the Methods section of the manuscript.

Comment 2. The conclusions section is missing

Response 2. The conclusions section was merged into the end of the discussion section, as it was not mandatory according to the journal policy to have a separate section for conclusions. However, in the corrected version of the manuscript, we have extracted the conclusions as a separate section.

Comment 3. Minor editing of  english lenguage required

Response 3. We spellchecked the manuscript and corrected some grammar and style issues

All changes in edited manuscript are highlighted in yellow

Reviewer 2 Report

Comments and Suggestions for Authors

Overall, the study addresses an important and understudied topic with potential implications for healthcare quality improvement. Elaborating on a few aspects of the background and methodology is needed for readers to gain a clearer understanding of how the study was conducted, ensuring transparency and reproducibility. Addressing these points will help strengthen the methodological rigor of the study and enhance its credibility within the scientific community. The following are specific comments:

1.     Abstract & background: Description of patient preparation aspects is lacking in both abstract and background. The significance of the various aspects of patient preparation -that are asked as questions- is important for background section for cohesiveness. Is there any published literature on how these aspects are associated with outcomes?

2.     Materials & methods: The methodology section lacks sufficient detail to ensure clarity regarding the design of the questionnaire, especially its development process and the items included. eg: Provide a detailed account of the development process of the questionnaire. Include information on how the questionnaire was designed, such as the identification of key areas or constructs related to patient preparation. Discuss whether any existing instruments were adapted or modified for the purposes of this study, and justify any modifications made. Explain the rationale behind the selection of specific items included in the questionnaire. How were these items determined to be relevant for assessing the attitude of non-laboratory healthcare professionals towards patient preparation? Were they derived from existing literature, expert opinions, or other sources? Specify how anonymity and confidentiality were ensured in the dissemination and collection of responses to mitigate potential biases.

3.      Results:

a.     the reported educational categories seem to be odd – given the respondents were nurses and physicians – what is a master’s degree one of the categories? For Physicians – why wasn’t a doctor of medicine or professional degree asked? Its surprising to see 0 physicians with professional bachelors or university degree – how can this information be interpreted?

4.     Discussion:

a.     Line 222-224: “Study findings demonstrate that, essentially, the opinions of both physicians and nurses align with existing research, evidence, and guidelines.” Which existing research, evidence and guidelines are being mentioned here? include these in the background section.

Author Response

Dear Reviewer,

Thank you very much for taking the time to review our manuscript titled "Assessing Non-Laboratory Healthcare Professionals' Attitude Towards the Importance of Patient Preparation for Laboratory Tests" We appreciate the thoughtful feedback and constructive comments you provided. Below, we address each of your points and outline the revisions we have made in response:

Comment 1. Abstract & background: Description of patient preparation aspects is lacking in both abstract and background. The significance of the various aspects of patient preparation -that are asked as questions- is important for background section for cohesiveness. Is there any published literature on how these aspects are associated with outcomes?

Response 1. We have enriched the abstract and background sections regarding the aspects of patient preparation, including several studies on the importance of patient preparation for the reliability of test results. Line 11-19 and Line 57-79, respectively.

Comment 2. Materials & methods: The methodology section lacks sufficient detail to ensure clarity regarding the design of the questionnaire, especially its development process and the items included. eg: Provide a detailed account of the development process of the questionnaire. Include information on how the questionnaire was designed, such as the identification of key areas or constructs related to patient preparation. Discuss whether any existing instruments were adapted or modified for the purposes of this study, and justify any modifications made. Explain the rationale behind the selection of specific items included in the questionnaire. How were these items determined to be relevant for assessing the attitude of non-laboratory healthcare professionals towards patient preparation? Were they derived from existing literature, expert opinions, or other sources? Specify how anonymity and confidentiality were ensured in the dissemination and collection of responses to mitigate potential biases.

Response 2. In the Methods section, we specified that the questionnaire was self-designed, so no existing tools adjusted. The questionnaire items were determined based on similar studies in the field and available international guidelines for venous blood sampling, particularly EFLM-COLABIOCLI. This information is also now included in the Methods section. Additionally, we provided further details on anonymity in the Methods subsection of Data Collection, stating, "Data were collected anonymously (no data regarding respondents' traceability was collected, nor did the researchers know which employees were invited)...".

Comment 3. Results. the reported educational categories seem to be odd – given the respondents were nurses and physicians – what is a master’s degree one of the categories? For Physicians – why wasn’t a doctor of medicine or professional degree asked? Its surprising to see 0 physicians with professional bachelors or university degree – how can this information be interpreted?

Response 3. In Lithuania, a master's degree typically follows the completion of a bachelor's degree and is awarded after the successful completion of a graduate program, which usually lasts for two years. The master's degree signifies a higher level of expertise and specialization in a particular field of study compared to the bachelor's degree. In Lithuania, physicians typically pursue a master's degree as part of their medical education. The medical education system in Lithuania follows the European model, where students complete a six-year program leading to a master's degree in medicine (MD). To avoid unnecessary interpretations, we clarified "master's degree" as "university master's degree" in the manuscript. Professional titles were collected from all respondents, who were then divided into two groups: nurses and physicians. Due to the low number of respondents per professional title, no analysis was conducted on this variable. This information is mentioned from Line 178 onwards.

Comment 4. Discussion: Line 222-224: “Study findings demonstrate that, essentially, the opinions of both physicians and nurses align with existing research, evidence, and guidelines.” Which existing research, evidence and guidelines are being mentioned here? include these in the background section.

Response 4. Included in the background section from Line 57.

All changes in edited manuscript are highlighted in yellow

Reviewer 3 Report

Comments and Suggestions for Authors

Comments on the Quality of English Language

Author Response

Dear Reviewer,

Thank you very much for taking the time to review our manuscript titled "Assessing Non-Laboratory Healthcare Professionals' Attitude Towards the Importance of Patient Preparation for Laboratory Tests" We appreciate the thoughtful feedback and constructive comments you provided. Below, we address each of your points and outline the revisions we have made in response:

Comment 1. KEYWORDS: Only pre-analytical phase is a U.S. National Library of Medicine Medical Subject Heading (MeSH) term. To facilitate searches for this manuscript, the authors are encouraged to identify MeSH terms that reflect meaning of the other keywords they have selected (eg, Clinical Laboratory Testing, Quality of Health Care)

Response 1. We have updated the keywords to align with the MeSH database. Line 43

Comment 2. Line 89: Data is the plural form of datum. Therefore, “Data were collected….”

Response 2. Corrected. Line 142

Comment 3. Lines 94-106: The authors compare the results from four different groups for two different independent variables (years in practice and years since last training)....

Response 3. In response to the reviewer's concern about the number of pairwise comparisons generated, we employed the Kruskal-Wallis test as our primary analysis method. This nonparametric test was chosen due to its ability to handle multiple independent groups without assuming normality, which aligns with the nature of our dataset. However, to address potential differences between specific groups, we conducted post-hoc pairwise comparisons using appropriate statistical adjustments, such as Bonferroni correction, to control for Type I errors.

Regarding the consideration of multiple nonparametric regression, it's worth noting that SPSS, the statistical software used for our analysis, does not inherently support this technique. As such, we focused on employing the Kruskal-Wallis test, which is well-supported by SPSS and widely recognized for its suitability in comparing multiple groups with non-normally distributed data. While multiple nonparametric regression could potentially offer additional insights, its implementation would require alternative statistical software or programming languages beyond the scope of our current analysis tools. Also, our primary aim was to ascertain whether significant differences existed between groups.Therefore, we believe our chosen approach with the Kruskal-Wallis test and post-hoc comparisons provides a robust analysis within the constraints of SPSS capabilities.

Comment 4. Line 117: Subject/verb disagreement. The subject of this sentence is plural; however, the verb is singular.

Response 4. Corrected. Line 183

Comment 5. Line 149: Please indicate which group (< 20 yrs experience or > 20 yrs experience) “scored lower” on Questions 5-7, 11. And rather than “Substantial disparities,” use, “Statistically significant differences.”

Response 5. Corrected as suggested. Line 216

Comment 6. Line 153: Entire comment regarding Line 149 applies here as well.

Response 6. Corrected as suggested. Line 227

Comment 7. Line 169: Please clarify which topic is being referred to.

Response 7. Sentence was clarified. Line 242

Comment 8. Lines 180-185: Entire comment regarding Line 149 applies here as well.

Response 8. Corrected as suggested. Line 249

Comment 9. Lines 195-205: Entire comment regarding Line 149 applies here as well.

Response 9. Corrected as suggested. Line 260-269

Comment 10. Line 209: As noted in the comment regarding Line 89, data is a plural word; therefore, “The data are….”

Response 10. Corrected. Line 273

Comment 11. Line 216: “… the results of the analysis of internal consistency….”

Response 11. Corrected as suggested. Line 281

Comment 12. Lines 219-220: “… offered insights into the attitudes and opinions of both nurses and physicians…” regarding what? The authors should restate the study’s objective here…

Response 12. The sentence has been rewritten to include the objective of the study for clarification. Line 285-286

Comment 13. Lines 220-222: This sentence reads rough and should be revised.

Response 13. The sentence has been rewritten. Line 286-289

Comment 14. Line 223: Recommend: “… the opinions of both physicians and nurses regarding the influence of the preanalytical variables on laboratory test results align with….”

Response 14. This sentence updated as reccomended. Line 290

Comment 15. Lines 231-234: The authors somewhat overstate their findings here. Their data show that among all respondents with less than 20 years of experience, a stronger consensus was attained regarding the impact of 4/12 preanalytical factors on lab test re....

Response 15. This section has been rewritten to ensure that the findings are not overstated. Line 299-304

Comment 16. Lines 235-237: Here, the authors mix and match the independent variables examined. The two independent variables in the study were Years of Practice and Years since Last Training. While physician respondents who have practiced for fewer tha...

Response 16. This section has been rewritten to ensure that possible attendance in training is not underestimated. Line 305-309

Comment 17. Lines 247-297: Several errors in English grammar and punctuation use occur in this section that require attention.

Response 17. This section of the manuscript has been revised, and grammar errors have been corrected.

Comment 18. In the conclusion, please specify the cutoff in Years of Experience and Years since Last Training. The authors also over-generalize their finding. The authors’ conclusions apply only to those non-laboratory healthcare professionals who responded to this particular survey.

Response 18. Conclusions rewritten as suggested. Refer to conclusions section.

Comment 19. Figures 1 and 2: Recommend revising the captions to, “Nurse-provided responses…,” and, “Physician-provided responses…,” respectively.

Response 19. Corrected as suggested.

Comment 20. Figures 3-6: Recommend revising captions to, “Comparison of attitudes regarding [or towards] the influence of….”

Response 20. Corrected as suggested.

All changes in edited manuscript are highlighted in yellow

Reviewer 4 Report

Comments and Suggestions for Authors

This review critically evaluates the article titled "Assessing Non-Laboratory Healthcare Professionals' Attitude Towards the Importance of Patient Preparation for Laboratory Tests" by Ricardas Stonys and Dalius Vitkus. The review covers several aspects as requested, including authenticity, language quality, methodology, novelty, data analysis, typographical errors, table quality, and bibliography currency and relevance.

The research addresses a genuine gap in healthcare literature regarding non-laboratory healthcare professionals' perceptions of patient preparation for laboratory testing. The study's design, from the survey creation to the analysis, shows an authentic effort to explore this under-researched area.

The methodology is robust and well-articulated, employing a specially designed questionnaire disseminated among a significant number of healthcare institutions. The statistical analysis, including Cronbach’s alpha for reliability and the Mann-Whitney U test for group comparisons, is appropriate for the data type and research questions. However, the article could benefit from a more detailed explanation of the questionnaire design process and the rationale behind the selection of specific statistical tests.

The study contributes new insights into an area that has not been extensively studied in Lithuania or broadly. It fills a significant gap by focusing on non-laboratory healthcare professionals' attitudes towards patient preparation, a critical aspect of laboratory testing that influences test outcomes.

The analysis is thorough, with a clear presentation of findings supported by descriptive statistics and inferential analysis. The division of respondents into meaningful groups based on experience and training adds depth to the analysis. Nevertheless, discussing the potential implications of these findings in a broader context would strengthen the impact of the study.

The document has a few minor typographical errors and inconsistencies, particularly in the formatting of references. A careful review and correction of these errors are recommended.

The tables are well-constructed, providing a clear and concise summary of the demographic characteristics and main findings. However, ensuring consistency in formatting and perhaps adding a few explanatory footnotes could enhance their utility.

The bibliography is current and relevant, with citations from reputable sources that are appropriate for the study's scope. The references support the study's premises and discussions effectively. Yet, the inclusion of more recent studies, if available, would be beneficial to demonstrate ongoing relevance.

This article presents valuable insights into the attitudes of non-laboratory healthcare professionals towards patient preparation for laboratory tests in Lithuania. The methodology is sound, the analysis is rigorous, and the findings contribute to the existing literature by highlighting the importance of proper patient preparation and the role of continuous training.

However, before acceptance, a revision is recommended to address the noted issues, including:

- Enhancing the clarity and flow of the language.

- Providing more detail on the questionnaire design and the choice of statistical methods.

- Expanding the discussion to place the findings in a broader context.

- Correcting typographical and formatting errors, especially in the references.

Comments on the Quality of English Language

The article is well-written in English, with minor issues. It maintains a professional tone and uses terminology appropriate for a scientific publication.

Author Response

Dear Reviewer,

Thank you very much for taking the time to review our manuscript titled "Assessing Non-Laboratory Healthcare Professionals' Attitude Towards the Importance of Patient Preparation for Laboratory Tests" We appreciate the thoughtful feedback and constructive comments you provided. Below, we address each of your points and outline the revisions we have made in response:

Comment 1. Enhancing the clarity and flow of the language.

Response 1. Language has been rechecked, and some changes have been made to improve the flow.

Comment 2. Providing more detail on the questionnaire design and the choice of statistical methods

Response 2. Added more details on how the questionnaire was designed in the Methods section. Also, additional information was included on why selected statistical methods were chosen.

Comment 3. Expanding the discussion to place the findings in a broader context.

Response 3. Expanded disccusion by discussing the potential implications of our findings. Line 322-330

Comment 4. Correcting typographical and formatting errors, especially in the references

Response 4. The manuscript was revised for typographical and formatting errors. Errors found were corrected, and references were revised according to the rules of the journal.

All changes in the manuscript are highlighted in yellow.

Reviewer 5 Report

Comments and Suggestions for Authors

Although not Earth-shaking, this is nonetheless a useful contribution to the literature.

We are told how many institutions were contacted about participating in this study and how many individuals responded, but this does not tell us the response rate. So, it is difficult to assess the representativeness of the sample.

It would be useful to say more about which medical staff are responsible for educating patients about how to prepare for lab tests. Does current practice agree with policy? Does this process include explaining the purpose of the test and obtaining the patient’s agreement to cooperate (if not full informed consent)?

Comments on the Quality of English Language

There are a few infelicitous or awkward sentences, but nothing that would inhibit understanding.

Author Response

Dear Reviewer,

Thank you very much for taking the time to review our manuscript titled "Assessing Non-Laboratory Healthcare Professionals' Attitude Towards the Importance of Patient Preparation for Laboratory Tests" We appreciate the thoughtful feedback and constructive comments you provided. Below, we address each of your points and outline the revisions we have made in response:

Comment 1. We are told how many institutions were contacted about participating in this study and how many individuals responded, but this does not tell us the response rate. So, it is difficult to assess the representativeness of the sample.

Response 1. Due to the recruitment process, it is challenging for us to calculate the response rate. We acknowledge this issue as a limitation of our study, as discussed at the end of the Discussion section. Line 355-358

Comment 2. It would be useful to say more about which medical staff are responsible for educating patients about how to prepare for lab tests. Does current practice agree with policy? Does this process include explaining the purpose of the test and obtaining the patient’s agreement to cooperate (if not full informed consent)?

Response 2. The discussion of this issue is slightly extended in the Background section. Line 82-91

All changes in the manuscript are highlighted in yellow.

Round 2

Reviewer 3 Report

Comments and Suggestions for Authors

The authors have adequately addressed my concerns. Line 144 in the revised manuscript still needs to treat "data" as a plural word throughout the sentence, “… no data regarding respondents’ traceability were collected...”.

This reviewer appreciates the authors' comments regarding their statistical analysis. The reviewer was unaware that SPSS V.22 could conduct post-hoc testing following a significant Kruskal-Wallis test, a capability that the reviewer was able to confirm via a Google search. Thank you for bringing this to the reviewer's attention.